# Extraction, Isolation and Characterization of Mycosporine-like Amino Acids from Four Species of Red Macroalgae

**DOI:** 10.3390/md19110615

**Published:** 2021-10-28

**Authors:** Yingying Sun, Xiu Han, Zhijuan Hu, Tongjie Cheng, Qian Tang, Hui Wang, Xiaoqun Deng, Xu Han

**Affiliations:** 1State Key Laboratory of Food Science and Technology, Jiangnan University, Wuxi 214122, China; 2Jiangsu Institute of Marine Resources Development, Jiangsu Ocean University, Lianyungang 222005, China; hanxiu163@163.com (X.H.); hzjd1230@163.com (Z.H.); ctj2007@126.com (T.C.); tang18974472526@163.com (Q.T.); X_uzhifan_77@163.com (H.W.); dxq2459260279@163.com (X.D.); hx1005x@163.com (X.H.); 3Jiangsu Key Laboratory of Marine Biotechnology, Jiangsu Ocean University, Lianyungang 222005, China; 4A Co-Innovation Center of Jiangsu Marine Bio-Industry Technology, Lianyungang 222005, China

**Keywords:** extraction, identification, isolation, mycosporine-like amino acids, red macroalgae

## Abstract

Marine macroalgae is known to be a good source of mycosporine-like amino acids (MAAs), especially red macroalgae. As a new type of active substance with commercial development prospects, the current progress in the extraction, isolation and characterization of MAAs is far from sufficient in terms of effectiveness in application. To determine the extraction processes of MAAs from four species of red macroalgae (*Bangia fusco-purpurea*, *Gelidium amansii*, *Gracilaria confervoides*, and *Gracilaria* sp.), a series of single-factor and orthogonal experiments were carried out in which the effects of solvents, the solid–liquid ratio, the time of extraction, the extraction degree and the temperature, on the yields of MAA extracts, were analyzed. Further, the isolation and identification of MAAs from *Bangia fusco-purpurea* and *Gracilaria* sp. were investigated. The results showed that the solid–liquid ratio, the time of extraction, the extraction degree and the temperature were 1:20 g/mL, 2 h, three times and 40 °C, respectively, when 25% methanol or 25% ethanol were used as the extraction solvent; these values were found to be suitable for the extraction of MAAs from four species of red macroalgae. Silica gel thin-layer chromatography was successfully used, for the first time, for the detection MAAs in this work, and it could be clearly seen that *Bangia fusco-purpurea* had the highest contents of MAAs among the four species of red macroalgae. MAA extracts from *Bangia fusco-purpurea* (or *Gracilaria* sp.) were isolated by silica gel column chromatography to obtain one fraction (or two fractions). The compositions and proportions of the MAAs in these fractions were determined via HPLC-ESI-MS spectra and by comparison with existing studies. Shinorine, palythine and porphyra-334 were found in 95.4% of the T_1_ fraction, and palythenic acid was found in 4.6% of this fraction, while shinorine, palythine and porphyra-334 were found in 96.3% of the J_1_ fraction, palythenic acid was found in 3.7% of the J_2_ fraction, and palythine was found in 100% of the J_2_ fraction, taken from the MAA extracts found in *Bangia fusco-purpurea* and *Gracilaria* sp., respectively. In addition, the relevant compositions and proportions of the MAA extracts taken from *Gelidium amansii* and *Gracilaria confervoides* were identified. This was the first study to report on the extraction process, isolation and identification of MAAs from *Bangia fusco-purpurea*, *Gelidium amansii*, *Gracilaria confervoides*, and *Gracilaria* sp.

## 1. Introduction

Mycosporine-like amino acids (MAAs) are UV-absorbing compounds with their main absorption maxima wavelengths between 310 nm and 360 nm [1]. The absorption maxima wavelength of some MAAs was found to be less than 310 nm [2] or greater than 360 nm [3]. Research on MAAs has undergone considerable development during the last 30 years [4,5,6,7,8,9,10,11], particularly in terms of their distribution [4,5,6], properties [4,6], and chemical characterization [4,7,8], in addition to other areas [9,10,11]. MAAs are a class of secondary metabolites with multiple activities due to their structural diversity [4,9]. Up to now, nearly 30 different MAAs have been identified [12]. They are widely distributed in marine red algae, cyanobacteria, phytoplankton and marine microorganisms [12]. As a producer of MAAs, red macroalgae are considered to be one the best sources of MAAs because of their high contents of numerous kinds of MAAs (Figure 1) [3,13]. 

It is difficult to obtain high-purity MAAs by extraction and isolation because of their strong water solubility. In Carreto and Carignan’s review [14], some methods of MAA isolation and identification were summarized, such as gel permeation [15,16], ion exchange resins [15,16,17,18], preparative TLC on silica gel [17], etc. Up to now, the extraction, isolation and purification of MAAs from red macroalgae have been investigated, for example, *Bostrychia scorpioides* [19], *Chondrus yendoi* [16,18], *Gelidium corneum* [6], *Gracilaria changii* [20], *Porphyra haitanensis* [21], *Porphyra tenera* [15], and other red macroalgae [22,23,24]. However, the research conducted thus far is not sufficient in terms of the application of MAAs from red macroalgae. The composition of MAAs is different in different species of red macroalgae, and thus, their appropriate extraction and isolation processes will be different. Therefore, it is necessary to study the extraction and isolation of MAAs from more species of red macroalgae for the development and application of MAAs.

*Bangia fusco-purpurea*, *Gelidium amansii*, *Gracilaria confervoides*, and *Gracilaria* sp., are red alga belonging to the orders Bangiales, Ceramiales, and Gracilariales, respectively. Red macroalgae belonging to these three orders have higher MAAs (Figure 1) according to our previous review [13]. The aim of this research was as follows: first, to establish the extraction process of MAAs from *Bangia fusco-purpurea*, *Gelidium amansii*, *Gracilaria confervoides*, and *Gracilaria* sp.; second, to carry out the isolation and characterization of MAAs. This is the first study on the extraction, isolation and identification of MAAs from these four species of red macroalgae. 

## 2. Results

### 2.1. Extraction Process of MAAs from Four Species of Red Macroalgae

#### 2.1.1. Extraction Solvents

MAAs are a class of water-soluble compounds, and are often used with polar solvents in extraction processes, such as aqueous methanol or ethanol. In this study, three different solvents (25% methanol, 25% ethanol, and distilled water) were used in the extraction of MAAs from *Bangia fusco-purpurea*, *Gelidium amansii*, *Gracilaria confervoides*, and *Gracilaria* sp. The results showed these three extraction solvents could be used in the extraction of MAAs from four red macroalgae (Figure 2). However, there were significant differences (*p* < 0.05) in the yield of MAA extracts obtained using different extraction solvents (Figure 2). For the extraction of MAAs from *Gracilaria* sp., the extraction solvent was 25% ethanol, which was able to obtain a higher yield of MAA extracts (19.99%~32.34%), while for the three other red macroalgae, higher yields of MAA extracts could be obtained (9.01%~12.39%) when the extraction solvent was 25% methanol. In addition, it is worth noting here that the yield of MAA extracts from *Gracilaria confervoides* was the lowest when 25% ethanol was used as an extraction solvent. These specific reasons need to be analyzed in the follow-up work. 

#### 2.1.2. Solid–Liquid Ratio, Time of Extraction, Extraction Degree and Temperature

The effects of solid–liquid ratio, time of extraction, and extraction degree on the yield of MAA extracts from four red macroalgae were similar; specifically, with the increasing of the factor levels, the yields of the MAA extracts increased, reached their maximum at a certain level, and then with the increase in the factor levels, the yields of the MAA extracts began to decline (Figure 3).

Except for MAA extracts from *Bangia fusco-purpurea*, the solid–liquid ratio had a significant (*p* < 0.05) effect on the yield of MAA extracts from other three macroalgae. When the solid–liquid ratio was 1:25, 1:20, 1:20, and 1:15 g/mL, respectively, the yields of MAA extracts from *Bangia fusco-purpurea*, *Gelidium amansii*, *Gracilaria confervoides* and *Gracilaria* sp. were the highest (Figure 3a). Since there was no significant difference (*p* > 0.05) in the yield of MAA extracts from *Bangia fusco-purpurea* (or *Gracilaria* sp.) when the solid–liquid ratio was between 1:25 g/mL and 1:20 g/mL (or 1:15 g/mL and 1:20 g/mL), the solid–liquid ratio could be uniformly determined as 1:20 g/mL in the subsequent extraction process of MAAs from four red macroalgae. 

In Figure 3b,c, the yields of MAA extracts from *Bangia fusco-purpurea*, *Gelidium amansii*, *Gracilaria confervoides* and *Gracilaria* sp. were the highest when the time of extraction (or temperature) was 3 h (or 40 °C), 2 h (or 40 °C), 2 h (or 45 °C) and 2 h (or 40 °C), respectively. However, for MAA extracts from *Bangia fusco-purpurea*, there was no significant (*p* > 0.05) difference in the yield of MAA extracts when the time of extraction was either 2 h or 3 h. Thus, the time of extraction was determined as 2 h for the extraction of MAAs from these four red macroalgae.

The extraction degree had significant effects (*p* < 0.05) on the yield of MAA extracts from *Bangia fusco-purpurea*, *Gracilaria confervoides* and *Gracilaria* sp. (Figure 3d). The yields of MAA extracts from these three red macroalgae, when extraction was conducted three times, were significantly (*p* < 0.05) higher than that when extraction was conducted one time. When the extraction degrees were increased to four times, the yields of MAA extracts did not increase significantly (*p* > 0.05). In the extraction process of MAAs from *Gelidium amansii*, the extraction degree had no significant (*p* > 0.05) effect. Therefore, for the extraction of MAAs from four red macroalgae, the most suitable extraction degree was three times. 

Further, the extraction conditions were optimized by orthogonal experiments. The results showed that the optimized extraction processes of MAAs from four red macroalgae were different, and that the effects of the four factors on the extraction of the MAAs were different (Table 1). In order to facilitate the subsequent extraction, the comprehensive balance method was used to obtain the unified extraction process of MAA extracts from four red macroalgae. Finally, the unified extraction process (40 °C, 2 h, three times, and 1:25 g/mL solid–liquid ratio) was determined. 

Summarily, the most suitable extraction process of MAA extracts from *Bangia fusco-purpurea*, *Gelidium amansii*, *Gracilaria confervoides*, and *Gracilaria* sp. was found under the following conditions: the temperature, time of extraction, extraction degree and solid–liquid ratio were 40℃, 2 h, three times and 1:20 g/mL, respectively, with 25% methanol or 25% ethanol used as an extraction solvent. MAA extracts from these four red macroalgae were prepared by optimizing the extraction process (Table 2 and Figure 4).

It was clear that the yields of the MAA extracts from *Bangia fusco-purpurea* and *Gracilaria* sp. were the highest of the four red macroalgae (Table 2). The absorbance of these MAA extracts, at 330 nm, from four species of red macroalgae was determined at the concentration of 0.5 g/L (MAA extracts). Approximate MAA contents in these MAA extracts were also calculated. It could be seen that *Bangia fusco-purpurea* and *Gracilaria* sp. had higher MAA contents. Thus, MAA extracts from *Bangia fusco-purpurea* and *Gracilaria* sp. were further isolated in subsequent work. These four MAA extracts from four species of red macroalgae had red or yellow appearances (Figure 4), indicating that there were some impurities, such as pigments, in these extracts. In subsequent work, these color substances need to be removed.

### 2.2. UV and TLC Detection of MAA Extracts from Four Red Macroalgae

Due to the absorption wavelength characteristics (310 nm~360 nm) of the MAAs, UV wavelength-scanning was found to be a very intuitive and convenient means of detecting them. As MAAs are a class of nitrogen-containing compounds, we attempted to use the thin-layer chromatography (TLC) determination method for the detection of these MAA extracts. In Figure 5, it can be seen that these MAA extracts possessed the characteristic absorption properties of MAAs and the color reactions (yellow or purple spots) of nitrogen-containing compounds on the TLC plates are also visible. In addition, the MAA extracts from *Gelidium amansii* showed different absorption peak shapes and a light purple strip (not particularly clear) was visible on the TLC plate, which can be further analyzed in a future study.

The combination of UV and TLC could quickly determine the existence of MAAs, and the spotted area on the TLC plate directly reflected the level of MAA content, which was very meaningful in terms of the detection of MAAs. In Figure 5, it can be seen that the spotted areas of the MAA extracts from *Gracilaria* sp. were larger than those of the MAA extracts from three other red macroalgae. Using this method, we also determined that MAAs existed in brown macroalgae *Undaria pinnatifida* and green macroalgae *Codium fragile* (data unpublished; previous research [25] pointed out that this macroalgae contained MAAs). In another research study, regarding the isolation process of MAA extracts from *Gloiopeltis fucatas* and *Mazzaella* sp., it was only mentioned that the TLC method was used to detect the preliminarily isolated components of seaweed extracts; however, there was no description of TLC method in this work [26]. Therefore, it can be said that there is no report on the detection of MAAs by TLC in the existing literature.

### 2.3. Isolation and Characterization of MAA Extracts from Bangia fusco-purpurea and Gracilaria sp.

Since TLC was suitable for the detection of MAAs, we tried to separate the MAA extracts from *Bangia fusco-purpurea* and *Gracilaria* sp. by silica gel (200–300 mesh) column chromatography (3.0 × 35 cm) with methanol/ethanol/distilled water (8:10:0.5, *v*:*v*:*v*) as an eluent, and several elution fractions were obtained (Figure 6).

Up to now, there have been only a few studies on the isolation of MAAs by silica gel column chromatography, such as the isolation of the MAAs of *Agarophyton chilense* [3], *Bostrychia scorpioides* [19], *Champia novae-zelandiae* [3], *Gloiopeltis fucatas* [26], *Mazzaella* sp. [26], *Pyropia plicata* [3]. One elution fraction, denoted as T_1_ (tube 5~tube 8, 0.073 g), was isolated from the MAA extracts of *Bangia fusco-purpurea*. The HPLC of T_1_ is listed in Appendix A. Two peaks were detected with retention times at 1.7 min and 6.6 min. The largest peak (1) had absorption maxima at 319, 333 and 334 nm with an additional maximum at 225 nm, and peak (2) had its maximum at 337 nm. In Appendix A, mass spectra are shown. The mass spectrum of the first peak in the HPLC spectrum revealed three singly charged ion peaks [M + H]^+^ at m/z 333.1, m/z 245.1, and m/z 347.0, while the second peak led to [M + H]^+^ at m/z 329.0. Based on the available information on the HPLC-ESI-MS spectrum, as well as by comparison with the published reports [6,27,28], the MAA composition of this fraction could be determined. The four MAAs found were shinorine, palythine and porphyra-334 (in 95.4% of the T_1_ fraction), and palythenic acid (4.6%) in T_1_ isolated from the MAA extracts of *Bangia fusco-purpurea* (Table 3 and Figure 7). 

Two elution fractions—J_1_ (tube 5, 0.022 g) and J_2_ (tube 9~tube 11, 0.047 g)—were obtained from the MAA extracts taken from *Gracilaria* sp. by means of silica gel column chromatography. Among them, J_1_ showed two peaks with retention times at 1.8 min and 6.6 min, respectively. The largest peak (1) had absorption maxima at 319, 333 and 334 nm with an additional maximum at 223 nm, and peak (2) had its maximum at 337 nm; J_2_ presented an absorption peak with a retention time at 1.7 min and an absorption maximum at 319 nm, with an additional maximum at 231 nm. In Appendix A, the mass spectra of J_1_ and J_2_ are presented. Similarly, J_1_ showed four singly charged ion peaks [M + H]^+^ at m/z 333.1, m/z 245.1, m/z 347.0 and m/z 329.0; J_2_ had a singly charged ion peak [M + H]^+^ at m/z 245.1. In addition, based on the available information on the HPLC-ESI-MS spectrum, and by comparison with the published data [6,27,28], four MAAs were determined: shinorine, palythine and porphyra-334 were found in 96.3% of the J_1_ fraction, and palythenic acid was found in 3.7% of J_1_ (Table 3); and palythine (100% of the J_2_ fraction) was determined in J_2_ (Table 3 and Figure 7).

In addition, the HPLC-ESI-MS spectra of the MAA extracts from another two red macroalgae were measured (see Appendix A). In the corresponding spectrum, the λ_max_ and molecular mass values had some similarity to the MAAs extracted from *Bangia fusco-purpurea* and *Gracilaria* sp., but there were obvious differences, which suggested that different MAAs existed in *Gelidium amansii* and *Gracilaria confervoides* (for example, they could be unknown MAAs or other substances that are not included in the published data [6,27,28]) (Table 3). 

## 3. Discussion

Red macroalgae is one of the natural sources of MAAs. In our previous study [13], the proportion of red macroalgae was found to be more than 80% among more than 570 species of marine macroalgae that contain MAAs. There were many kinds of MAAs in red macroalgae. In order to develop these MAA sources, it was necessary to study the extraction process of MAAs from red macroalgae. Unfortunately, there are still very few studies about the extraction of MAAs from red macroalgae. So far, the extraction processes of MAAs from some red macroalgae, such as *Eucheuma* sp. [30,31], *Gloiopeltis furcata* [32], *Gracilaria chilensis* [33], *Porphyra* sp. [34], *Porphyra haitanensis* [21] and *Porphyra yezoensis* [24], have been reported. The extraction processes of MAAs from *Bangia fusco-purpurea*, *Gelidium amansii*, *Gracilaria confervoides*, and *Gracilaria* sp. have not yet been studied. There four red macroalgae belong to the orders Bangiales, Ceramiales, and Gracilariales, respectively. Figure 1 shows that the red macroalgae belonging to these three orders have higher MAAs. Therefore, these four species of red macroalgae were selected as the research objects in this work.

First, different solvents, solid–liquid ratios, times of extraction, extraction degrees and temperatures were analyzed in terms of their effects on the yields of MAA extracts from the four red macroalgae in this work (Figure 2 and Figure 3). Some research studies showed that the most common solvents were aqueous solutions of methanol or ethanol with different concentrations [9,24,35,36,37], and even distilled water [38], in the extraction process of MAAs. It was easy to conclude that the suitable solvents for the extraction of MAAs are different because the compositions of MAA extracts vary with different species of marine macroalgae [6,35,36,37,38]. In the extraction of MAAs from red alga dulse in the work of Usujiri, distilled water was more suitable as an extraction solvent [37]. For the extraction of MAAs from four red macroalgae (*Agarophyton vermiculophyllum*, *Crassiphycus corneus*, *Gracilariopsis longissima*, and *Porphyra leucosticta)*, there were no significant differences between total MAA concentrations using 20% aqueous methanol or distilled water as extraction solvents [38]. In our work, 25% methanol (or 25% ethanol) was a more appropriate extraction solvent for MAAs from *Bangia fusco-purpurea*, *Gelidium amansii* and *Gracilaria confervoides* (or *Gracilaria* sp.) (Figure 2). Of course, there were many other solvents for the extraction of MAAs, such as acetonitrile [39], 0.2% aqueous acetic acid with 0.5% methanol [40], 2-octyl dodecanol [23], etc. Temperature seems to be a neglected factor that affects the extraction of MAAs, and the most common temperatures ranged from 4 [23,25,38] to 45 ℃ [6,25,41]. In the extraction process of MAAs from *Gelidium amansii* and *Gracilaria* sp., the yields of MAA extracts increased significantly when the temperature was increased from 35 ℃ to 40 ℃ or 45 ℃. However, the yields of MAA extracts did not change significantly at other temperatures; for *Bangia fusco-purpurea* and *Gracilaria confervoides*, temperature also had no significant effect on the yields of MAA extracts (Figure 3d). The solid–liquid ratio, the time of extraction and the extraction degree were often considered in the extraction process of active substances, and they were also key factors for the extraction of MAAs from red macroalgae. Similarly, in some existing reports [19,28,29,30,31,32,33], it was found that larger solid–liquid ratios, longer extraction times and greater extraction degrees could obtain higher yields of MAAs. In this work, the effects of these three factors on the extraction of MAAs from four red macroalgae were similar to these reports [19,28,29,30,31,32,33]. It could be clearly seen that the effects of the solid–liquid ratio and extraction degree on the yields of MAA extracts from the four red macroalgae were more significant than that of the time of extraction in this work (Figure 3a–c). In the extraction of MAAs from marine macroalgae, the time of extraction has been found to be a factor with a wide range of values, ranging from more than ten minutes [19] to several hours [6] or over ten hours [40]; this is probably related to the position of MAAs within seaweed tissue. However, there no research has been conducted on the specific location of MAAs in marine macroalgae tissues. The relationship between the time of extraction of MAAs and the location of MAAs in different marine macroalgae is the only relevant area of speculation regarding this topic to date. Finally, through a series of orthogonal experiments (Table 1), the suitable extraction processes of MAA extracts from four red macroalgae were obtained, and the MAA extracts were prepared. It could be clearly seen that these extracts were mostly red or yellow (Figure 4); these colors were easy to remove in the subsequent isolation process using silica gel column chromatography. In summary, the optimum extraction process of MAA extracts from *Bangia fusco-purpurea*, *Gelidium amansii*, *Gracilaria confervoides*, and *Gracilaria* sp. is as follows: 25% methanol or 25% ethanol (20 times the volume of the macroalgae powder) is added into the materials, and extracted three times at 40℃, for 2 h each time. The yield (%) of five Russian seaweeds extracts using 50% aqueous ethanol as an extraction solvent [26] was significantly lower than that of the MAA extracts from the four species of marine macroalgae in our work.

In this study, we did not provide the exact content of MAAs in the MAA extracts from each marine macroalgae, but we gave their approximate values (Table 2). Therefore, it can be clearly seen that total contents of MAAs differed significantly between the four species of marine macroalgae. The MAA contents in *Gracilaria* sp. were the highest, followed by *Bangia fusco-purpurea*, and the lowest were found in *Gracilaria confervoides* and *Gelidium amansii* (Table 2). According to our previous study [13], the MAA contents of most red macroalgae (more than 60%) are between 1 and 2 mg/g among the 323 red macroalgae that contain MAAs. In the present study, approximate MAA content in *Gracilaria* sp. and *Bangia fusco-purpurea* was 1.532 and 2.645 mg/g, respectively (Table 2). It showed that these two red macroalgae were good sources of MAAs. UV wavelength-scanning of MAA extracts showed that they had maximum absorption between 310 and 360 nm, except for the MAA extracts from *Gelidium amansii*, for which the maximum absorption was less than 240 nm with obvious absorption between 260 and 280 nm and between 310 and 360 nm (Figure 5). The shape of these spectra indicated the presence of more than one MAA, which was confirmed in a later study about MAA identification in several fractions isolated from MAA extracts taken from *Bangia fusco-purpurea* and *Gracilaria* sp. (Table 3). In this work, TLC, in combination with UV wavelength scanning, was used to detect MAAs (Figure 5). This is a qualitative detection method. In previous studies [28,30,42], infrared spectroscopy [28], gas chromatography-mass spectrometry [43] and nuclear magnetic resonance spectroscopy [42] were noted as commonly used qualitative methods for MAAs. Although the sensitivity of TLC was not particularly high compared with IR, among others, it was sufficient for the detection of nitrogen-containing compounds, and the content levels of the MAAs in the tested sample could generally be directly compared via analysis of the spotted area in the TLC plate (Figure 5). This was the first report on the use of TLC to detect MAAs from *Bangia fusco-purpurea*, *Gelidium amansii*, *Gracilaria confervoides*, and *Gracilaria* sp.

Further, MAA extracts from *Bangia fusco-purpurea* and *Gracilaria* sp. were repeatedly isolated through silica gel column chromatography to obtain several elution fractions, namely T_1_, J_1_ and J_2_ (Figure 6). In some research studies [6,7,15,16,17,18,22,23,29,34,42,43], several methods, such as ion exchange column chromatography [6,15,16,17,18,22,23,34], HPLC [6,7,29], C18 solid phase extraction column [42], and hydrophilic interaction liquid chromatography (HILIC) [43], in addition to others [15,16], were used to separate MAAs from marine macroalgae. Less frequently, silica gel column chromatography was used to preliminarily isolate MAAs [3,19,27]. Preparative TLC was used on silica gel for the isolation of MAAs from the red-tide dinoflagellate *Alexandrium excavatum* [17]. In the aforementioned study, the MAAs were found to be colorless; however, in this work, the MAA extracts from four red macroalgae showed some colors because they contained some pigments, but steps were taken to remove these pigments in the extraction process. The pigments and MAAs were effectively separated by silica gel column chromatography. Some colors appeared in the first few tubes, and some colors were adsorbed at the top of the silica gel column after elution (Figure 6). In short, several isolated fractions (T_1_, J_1_ and J_2_) that contained MAAs were colorless after silica gel column chromatography. In de la Coba et al.’s research [6], porphyra-334 plus shinorine was isolated from the red alga *Gelidium corneum*, and shinorine was isolated from the red alga *Ahnfeltiopsis devoniensis*, by adsorption and ionic exchange chromatography. In sum, these works provide a good set of references for us to continue to purify the fractions T_1_ and J_1_.

To date, more than 20 kinds of MAAs have been identified in marine macroalgae [12,13], thus providing a lot of reference information for the identification of MAAs. In the present investigation, the major MAAs from *Bangia fusco-purpurea* and *Gracilaria* sp. were analyzed using UV, HPLC and MS spectra. Four kinds of MAAs were identified, namely, shinorine, palythine, porphyra-334 and palythenic acid (Table 3 and Figure 7). Shinorine, porphyra-334, and palythine were the most common MAAs described in the work of Bangiales and Gracilariales [13,44]; palythenic acid was found less frequently, although it was previously described for *Solieria chordalis*, which was the first seaweed to exhibit palythenic acid [8], and this MAA was found to be common in microalgae and phytoplankton [40]. The compositions of MAA extracts from *Gelidium amansii* and *Gracilaria confervoides* were also analyzed by HPLC and MS spectra. Some unknown MAAs or other substances were also found in their extracts (Table 3). To our knowledge, except the composition of MAAs from *Gelidium amansii*, which was previosuly reported [45], the compositions of MAAs from the other three species of red macroalgae were pointed out for the first time in this study. Nakamura et al. only determined that asterina-330, palythine and shinorine existed in *Gelidium amansii* [46]. However, porphyra-334 and gadusol (the precursor of MAAs) have not been identified in this red macroalgae. In our work, porphyra-334 and gadusol were found in *Gelidium amansii*, which constitutes a very useful addition to the subject

Red algae are one of the largest algae groups, with about 5000–6000 species, mainly comprising multicellular and marine macroalgae. In China, there are more than 1200 species of red macroalgae [46]. However, there are less than 20 species of red macroalgae for which clear MAA extraction and isolation processes are known [7,20,21,22,23,32,33,34,35,36,43]. A great deal of further work in this area needs to be carried out by researchers.

## 4. Materials and Methods

### 4.1. Materials

The dry red macroalgae were purchased by Jiangsu Bilian Marine Biotechnology Co., Ltd. (Jiangsu, China). These materials were cleaned, cut into small pieces, freeze-drying carried out, and ground into powder before the experiments.

### 4.2. Extraction of MAAs

#### 4.2.1. Single Factor Experiments

(1) Extraction solvent

Ten grams of dry powder were extracted with 200 mL of extraction solvent (25% methanol, 25% ethanol, or distilled water) for 2 h in 400 mL centrifuge tube and incubated in a water bath at 45 °C. After pouring out the extracts, extraction solvent was again added to the 400 mL centrifuge tube, and the extraction process was repeated once according to the above conditions. These extracts were combined, filtrated and evaporated under reduced pressure to obtain the concentrate. Absolute ethanol (final content of 80%) was added to the concentrate, and placed at −20 °C for 6 h. After centrifugation at 7000 g for 10 min at 4 °C, the supernatants were poured out. The precipitate was washed with distilled water for 2~3 times. The washing solution and the previous supernatants were combined, concentrated under reduced pressure at 45 °C, and freeze-drying was carried out. In this way, the MAA extracts were prepared. During the experimental process, three parallel samples were set. The calculation of the yield of MAAs was as follows: 

Yield of MAA extracts (%) = 100%×the quality of MAA extracts (DW, g)/ the quality of red macroalgae powder (DW, g). 

(2) The solid–liquid ratio, the time of extraction, the extraction degree and the temperature

The extraction process was carried out according to the method described above. the level of experimental factors was set according to Table 4. In the experiments conducted to determine the extraction temperature, the time of extraction, extraction degree and solid–liquid ratio were set to 2 h, 2 times and 1:20 g/mL, respectively. In the experiments conducted to determine the time of extraction, the extraction temperature, extraction degree and solid–liquid ratio were 45 °C, twice and 1:20 g/mL. In the experiments conducted to determine the extraction degree, the extraction temperature, the time of extraction and the solid–liquid ratio were 45 °C, 2 h and 1:20 g/mL. In the experiments conducted to determine the solid–liquid ratio, the extraction temperature, the time of extraction, and the extraction degree were set at 45 °C, 2 h and twice. Three parallel samples were also set. 

#### 4.2.2. Orthogonal Experiments

L_9_ (3^4^) was selected to further analyze the effects of the extraction temperature, the time of extraction, the extraction degree and the solid–liquid ratio on the extraction of MAAs from four species of red macroalgae (Table 5). In this process, parallel samples were not set.

#### 4.2.3. Approximate Quantification of MAAs

Ten grams of dry powder were extracted according the optimum extraction process. Furthermore, quantities of 0.005 g of the MAA extracts were dissolved in 10 mL of distilled water, filtrated, and the absorbance was determined at 330 nm. This process was undertaken with reference to the method reported by Tsujino et al. [18]; the approximate quantification of MAA contents in MAA extracts was conducted using an average molar extinction coefficient of 40,000 [18] and an average molecular weight of around 300 [45], and the results were expressed as mg/g DW.

### 4.3. Isolation of MAAs

Quantities of five grams of the MAA extracts were loaded in preparation for silica gel column chromatography (3.0 cm × 25 cm, 200~300 mesh), were eluted using a mixture containing methanol, ethanol and distilled water (8:10:0.5, *v*:*v*:*v*) at a quantity of 2.5 times the column volume, and received 50 mL of eluted fraction per tube. The elution rate was 1.0× the column volume/h. After the vacuum concentrations were determined, several fractions were obtained.

### 4.4. Dedection of MAAs

#### 4.4.1. Ultraviolet Spectrum

A quantity of 0.005 g of the sample (MAA extracts or isolated fractions) was dissolved in 10 mL of distilled water, filtrated using a 0.45 μm membrane filter, and scanned at wavelengths ranging from 200 to 400 nm in a UV spectrophotometer to determine the characteristic absorption of the sample.

#### 4.4.2. Silica Gel Thin Layer Chromatography

The sample was dissolved in distilled water (0.5 g/L), spotted on a silica gel G plate and developed, with a mixture of methanol, ethanol and distilled water (8:10:0.5, *v*:*v*:*v*) used as the developing agent. After the silica gel G plate was blown dry, the potassium iodide reagent (3.6 g bismuth potassium iodide, 10 mL glacial acetic acid, 60 mL distilled water) was sprayed and left to stand for 15~20 min. The presence of yellow or orange spots indicated that a positive MAA reaction had occurred.

#### 4.4.3. High Performance Liquid Chromatography with Diode-Array Detection (HPLC-DAD)

The sample was dissolved in distilled water (0.5 g/L, isolated fractions or MAA extracts), passed through a 0.22 μm membrane filter, and analyzed by HPLC-DAD (Waters) as follows: 0.2% formic acid aqueous solution and 0.2% formic acid methanol solution were used as mobile phase A and B, respectively. Elution was carried out at the following gradient: 0~20 min; B%: 0~70%; flow rate: 1.0 mL/min, temperature: 25 °C; 100 μL of the sample was injected into HSS T3 column (Waters, 150 mm × 4.6 mm, 3.5 μm). Peaks were detected at 330 nm and absorption spectra were recorded between 200 and 400 nm for each peak that was detected.

#### 4.4.4. HPLC-ESI-MS Spectrometry

As previously described [34], the sample was analyzed by liquid chromatography/tandem mass spectrometry (LC-MS^n^) using a HPLC System. The scanning mode was in positive ionization mode, and yielded spectra with the charges/masses (m/z) of the eluded peaks. The parameters of the mass spectrometer were as follows: nitrogen was used as a nebulizing (45 psi) and drying gas (10 L/min, 350 °C), the breaking voltage was 100 V, the capillary voltage was 4500 V, and full scan was used (Scan). The reference ion mass ratios were 121.0509 and 922.0098, which were real time corrections of the measurement results obtained for the reference ions. The resolution m/z at 922.0098 was 11,300 for the full scan, and the mass to charge ratio (m/z) range was 120 to 1000. MAAs were identified based on their maximum absorption peaks (λ_max_ nm), the mass to charge ratio (m/z) and the fragmentation pattern [M + H]^+^ in the second-order mass spectrum via comparison with published research results [6,28,29,47,48].

### 4.5. Data Treatment

The experimental data were tested and statistically analyzed using the SPSS11.5 software package. *p* < 0.05 was the threshold for statistical significance, and *p* < 0.01 indicated extreme statistical significance.

## 5. Conclusions

This study was the first to establish a set of suitable conditions for extracting MAAs from four species of red macroalgae (*Bangia fusco-purpurea*, *Gelidium amansii*, *Gracilaria confervoides*, and *Gracilaria* sp.). Shinorine, palythine, porphyra-334, and palythenic acid were determined for the first time in *Bangia fusco-purpurea* and *Gracilaria* sp. through isolation using silica gel column chromatography, identification via HPLC-ESI-MS spectra, and comparison with existing studies. In addition, an MAA palythine monomer was prepared from *Gracilaria* sp. This study provides a good technical reference for the isolation and purification of MAAs from marine macroalgae.

## Figures and Tables

**Figure 1 marinedrugs-19-00615-f001:**
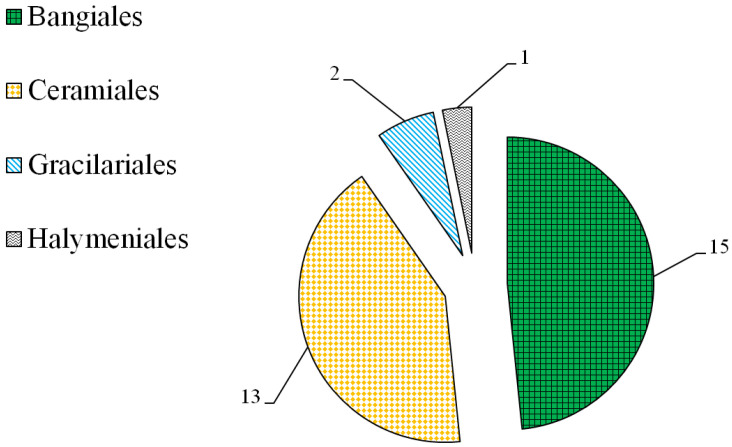
Ordered distribution of MAA contents present at >3 mg/g in red macroalgae (data from reports collected in Web of Science, Springer, Google Scholar and CNKI databases from 1999 to 2019). The numbers in the pie chart represent the total number of red macroalgae species belonging to this phylum.

**Figure 2 marinedrugs-19-00615-f002:**
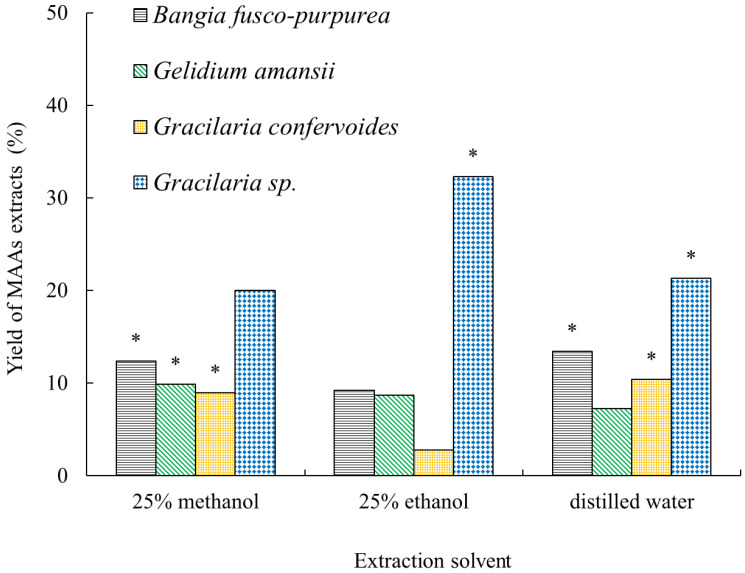
Effects of three solvents on the yield of MAA extracts from 4 red macroalgae. The data in the figure are the average of three parallel samples. * represents significant difference between the data, specifically *p* < 0.05.

**Figure 3 marinedrugs-19-00615-f003:**
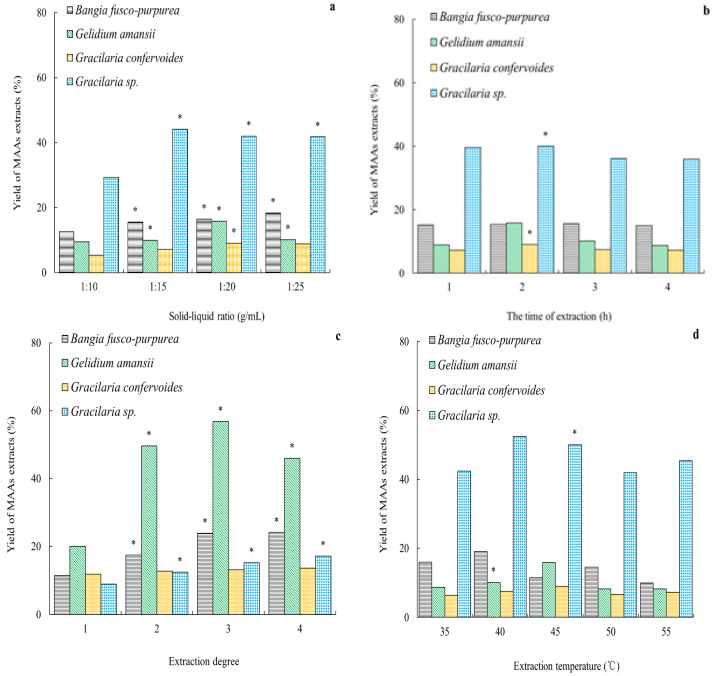
Effects of different factors on the yield of MAA extracts from 4 red macroalgae. The data in the figure are the average of three parallel samples. * represents significant difference between the data, specifically *p* < 0.05. ((**a**) Solid–Liquid Ratio, (**b**) Time of Extraction, (**c**) Extraction Degree and (**d**) Extraction Temperature)).

**Figure 4 marinedrugs-19-00615-f004:**
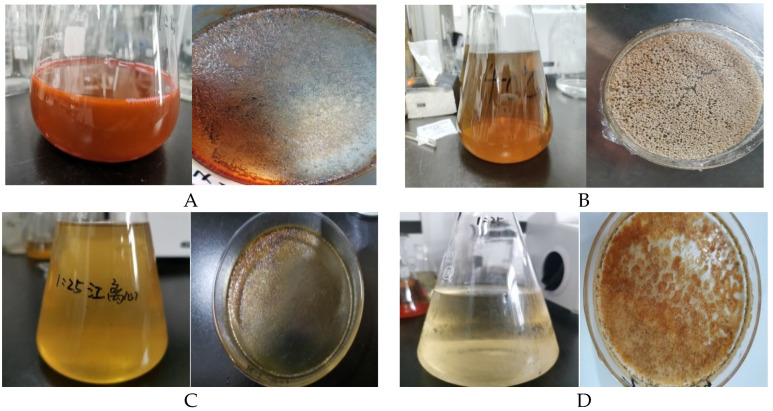
MAA extracts from four red macroalgae: (**A**) MAA extracts (concentrated solution and lyophilized powder) from *Bangia fusco-purpurea*; (**B**) MAA extracts (concentrated solution and lyophilized powder) from *Gelidium amansii*; (**C**) MAA extracts (concentrated solution and lyophilized powder) from *Gracilaria confervoides*; (**D**) MAA extracts (concentrated solution and lyophilized powder) from *Gracilaria* sp.

**Figure 5 marinedrugs-19-00615-f005:**
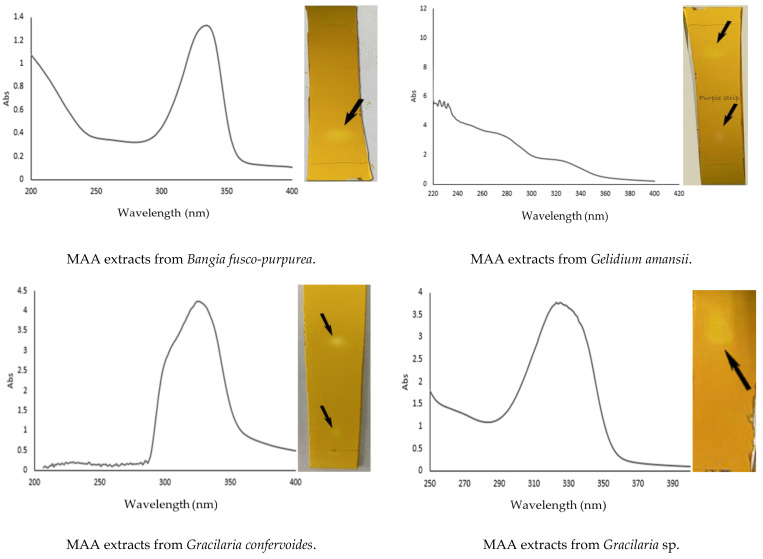
Wavelength scanning and TLC detection of MAA extracts from four red macroalgae.

**Figure 6 marinedrugs-19-00615-f006:**
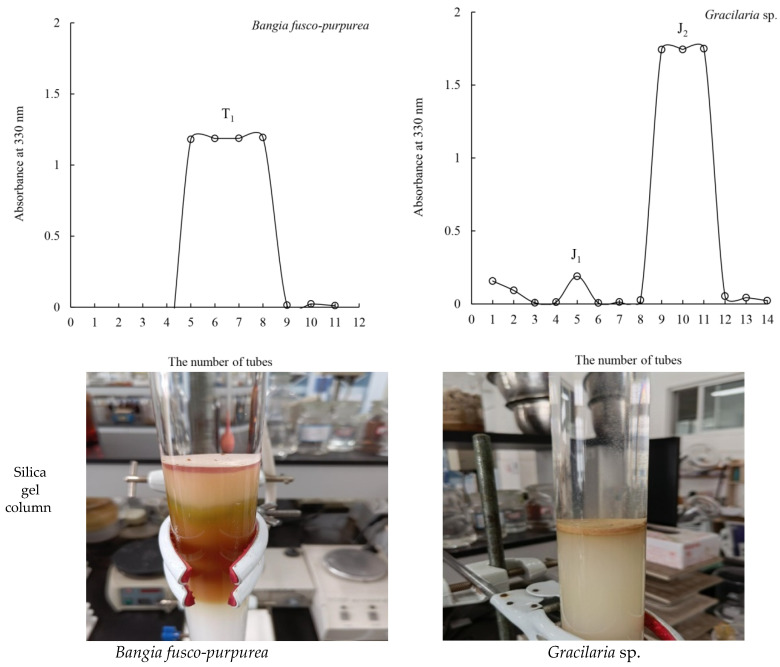
Isolation of MAA extracts from *Bangia fusco-purpurea* and *Gracilaria* sp. through silica gel column chromatography.

**Figure 7 marinedrugs-19-00615-f007:**
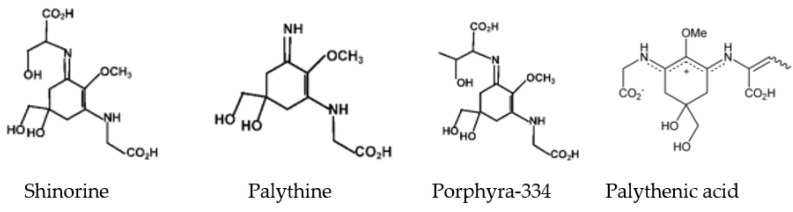
Structure of several MAAs.

**Table 1 marinedrugs-19-00615-t001:** Orthogonal experiments results.

No.	Factors				Quality of MAA Extracts (mg)			
	Temperature (°C) A	Time (h) B	Times C	Solid–Liquid Ratio (g/mL) D	*Bangia fusco-purpurea*	*Gelidium amansii*	*Gracilaria confervoides*	*Gracilaria* sp.
1	1	1	1	1	162.49	101.25	60.26	401.45
2	1	2	2	2	168.34	110.22	80.96	414.85
3	1	3	3	3	171.67	123.48	98.13	427.96
4	2	1	2	3	175.48	109.46	78.45	422.59
5	2	2	3	1	165.34	115.36	82.77	431.25
6	2	3	1	2	159.48	95.31	83.04	400.27
7	3	1	3	2	163.52	120.66	94.16	408.36
8	3	2	1	3	169.38	89.64	75.38	426.33
9	3	3	2	1	158.55	100.82	81.19	399.89
The order of factors					D > B > A > C	C > A > B > D	C > B > D > A	D > B > C > A
The optimizing combination					A_1_B_2_C_2_D_3_	A_1_B_1_C_3_D_2_	A_3_B_3_C_3_D_2_	A_2_B_2_C_3_D_3_
The unified extraction process					A_1_B_2_C_3_D_3_			

**Table 2 marinedrugs-19-00615-t002:** The yield (%, average value ± SD) of MAA extracts from four red macroalgae.

	*Bangia fusco-purpurea*	*Gelidium amansii*	*Gracilaria confervoides*	*Gracilaria* sp.
Yield (%)	21.81 ± 3.42	13.13 ± 2.26	14.96 ± 2.73	47.50 ± 3.55
Absorbance at 330 nm	1.021	0.619	0.665	1.763
Approximate MAAs content (mg/g)	1.532	0.929	0.998	2.645

Note: The yield was expressed as a percentage and calculated using the MAA extracts’ weight in proportion to the weight of the dry macroalgae powder. Absorbance and approximate MAAs content were the average of three parallel samples.

**Table 3 marinedrugs-19-00615-t003:** Information (maximum absorption wavelength, MS values, relative peak area and composition) on the isolated fraction or MAA extracts from four red macroalgae.

Fraction or Extracts	λ_max_ (nm)	[M + H]^+^	Mass	MAA	Proportion (%)
T_1_ (isolated from MAA extracts from *Bangia fusco-purpurea*)	333 319 334 337	333.1 245.1 347.0 329.0	332 244 346 328	Shinorine Palythine Porphyra-334 Palythenic acid	95.4 (Mixture of 3 MAAs) 4.6
J_1_ (isolated from MAA extracts from *Gracilaria* sp.)	333 319 334 337	333.1 245.1 347.0 329.0	332 244 346 328	Shinorine Palythine Porphyra-334 Palythenic acid	96.3 (Mixture of 3 MAAs) 3.7
J_2_ (isolated from MAA extracts from *Gracilaria* sp.)	319	245.1	244	Palythine	100
Extracts from *Gelidium amansii*	268 332 319 332 330 328	206.0 333.1 245.1 347.0 / 222.7	204 332 244 346 / 221	Gadusol [29] (the precursor of MAAs) Shinorine Palythine Porphyra-334 Unknown MAAs or other substances Unknown MAAs or other substances	/ 49.6 16.6 3.45 23.6 6.34
Extracts from *Gracilaria confervoides*	268 333 319 334 331 336 330	206.0 333.1 245.1 347.0 / 329.0 /	204 331 244 346 / 328 /	Gadusol Shinorine Palythine Porphyra-334 Unknown MAAs or other substances Palythenic acid Unknown MAAs or other substances	0.87 7.54 42.3 5.62 39.0 2.93 1.80

**Table 4 marinedrugs-19-00615-t004:** Factors and levels of single factor experiments.

Factors	Levels				
	1	2	3	4	5
Temperature/°C	35	40	45	50	55
Time of extraction/h	1	2	3	4	
Extraction degree	1	2	3	4	
Solid–liquid ratio/g/mL	1:10	1:15	1:20	1:25	

**Table 5 marinedrugs-19-00615-t005:** Factors and levels of orthogonal experiments.

Level	Factors			
	Temperature/°C	Time of Extraction/h	Extraction Degree	Solid–Liquid Ratio/g/mL
1	40	1	2	1:15
2	45	2	3	1:20
3	50	3	4	1:25

## Data Availability

The datasets used or analyzed during the current study are available from the corresponding author on reasonable request.

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
