# Peer review of "Extraction, Isolation and Characterization of Mycosporine-like Amino Acids from Four Species of Red Macroalgae"

_marinedrugs, 2021, doi:10.3390/md19110615_

Round 1

Reviewer 1 Report

As I understand it, the study aims to establish a set of conditions suitable for extraction of MAAs from a range of seaweeds by applying an orthogonal design to assess several parameters. From the results and discussion I can see that a method was specified that would obtain some MAAs from all four seaweeds.  However, I am not clear to what extent the method chosen enriches selectively for particular MAAs.  It would be useful to include more data to address this.  Aside from this, the study is a useful addition to the body of literature on MAAs in seaweeds.

The standard of English is sufficient to understand the study and results, with effort and care on the part of the reader.  However, it falls short of the fluency that would be expected for a published scientific article.  Some of the terminology is confusing, for example "extraction times" is used to refer to the number of successive extractions performed, and "extraction time" is used to refer to the length of the extraction.  The manuscript should be edited for English usage by a professional editor or native English speaker.

The figures are not easy to interpret.  In some cases they lack important information such as full legends that describe each feature.  The shading boxes in the legends are too small to see.  The meaning of asterisks are not explained.

Author Response

Dear Prof., Ph.D.,

I would like to thank the referees for your nice comments on our manuscript. I try my best to revise the manuscript with my co-authors. It spent me almost one week to contact with my co-authors and to discuss the improvement in both English grammar and contents of this manuscript. As you see, we made a major revision for this manuscript. You can find the modified and revised parts (marking out with red).

  1. MAAs were a class of water-soluble compounds, therefore, polar solvents (25% methanol or 25% ethanol) were used in the extraction process of MAAs from marine macroalgae in this work. Such an extraction solvent can ensure MAAs are extracted from marine macroalgae. at the same time, some water-soluble substances are also extracted, such as polysaccharides, proteins and other substances (they are uniformly called non-target substances). In the subsequent processing, these non-target substances are removed to the greatest extent through ethanol precipitation, low-temperature standing and centrifugation. In this way, MAAs extracts were prepared.
  2. Yes, "extraction time" and "extraction times" are very confusing. So, we modify them to "the time of extraction " and "extraction times".
  3. Yes, the comments are very nice. Some important information has been added now. And we revised several figures (Figure 2 and Figure 3) to make the legend clearer.

    We hope you could provide us more comments so that we can improve our future work. And I hope that our manuscript have chance to get a rapid publication in this Journal. Please understand my condition because the rapid publication of my work is important for me and our group. I am looking forward to your good news.

    Thank you for your attention and comments.

    Best Regards

    Yours sincerely

    Yingying Sun

Reviewer 2 Report

The comments of the manuscript "Extraction, isolation and characterization of Mycosporine like amino acid from 4 species of red macroalgae by Su et al are enclosed in the pdf files 

Author Response

Dear Prof., Ph.D.,

I would like to thank the referees for your nice comments on our manuscript. I try my best to revise the manuscript with my co-authors. It spent me five days to contact with my co-authors and to discuss the improvement in both English grammar and contents of this manuscript. As you see, we made a major revision for this manuscript. You can find the modified and revised parts (marking out with red).

  1. Yes, this is a good suggestion. In the discussion part, we compare this research results with those of our paper.
  2. Yes, this is a very good suggestion. In the introduction part, we quote and discuss the corresponding technology in this review (Carreto and Carignan, 2011).

  3. Thank you very much for your suggestion. We checked the data again and found “quality” represent quality of MAAs extracts, are not MAAs contents in the original table 2. In order not to be confused, we deleted a line “Quality” and replaced it with “330 nm absorbance” line.

  4. But as experts say, the color is not a precise indicator of the quality of the MAAs extraction. In MAAs extraction process, we don't use color as a measure of MAAs extraction. In this paper, the color of MAAs is mentioned in two places, one is the detection of MAAs using TLC, and the other is prepared MAAs extracts. The former only determines the presence of MAAs by color, and the latter indicates that there are impurities in MAAs extracts by color. In order to express it more clearly, we have supplemented it in the corresponding paragraph.
  5. Yes, you're right. The content of gadusol is very low. In their HPLC chromatogram (in the supplementary materials), the peak area value of the corresponding peak was less than 1.0%. And now it has been added. In HPLC chromatogram of Gelidium amanssi, the peak area corresponding to gadusol was not determined.
  6.  You're quite right. MAAs extracts have probably coumarins or coumarins derivatives. We immediately conducted an experiment and found that the MAAs extract showed fluorescence under the UV analyzer. So, we modified the corresponding expression to present the research results more accurately. 
  7. Yes, this is a very good suggestion. We made corresponding additions in the discussion section. However, since our results did not calculate MAAs concentration, we did not compare the data. 

  8.  This is our clerical error, which has been corrected now.

  9. The original intention of establishing this database is to provide convenient query ways for researchers all over the world. However, because the database uses the campus network, which limits the authority of external query, it can not be accessed at present. We are trying to restore the authority of query, but it will take some time because we need to coordinate multiple departments.

    We hope you could provide us more comments so that we can improve our future work. And I hope that our manuscript have chance to get a rapid publication in this Journal. Please understand my condition because the rapid publication of my work is important for me and our group. I am looking forward to your good news.

    Thank you for your attention and comments.

    Best Regards

    Yours sincerely

    Yingying Sun

Reviewer 3 Report

The manuscript describes new experimental methods of extraction of mycosporine-like amino acids from 4 different Macroalgae species.

The current version of the manuscript is not suitable for publication in a scientific peer reviewed journal.

Major revisions:

-Extensive English revision

-More logic and concise description of the experimental work

-The structures of the isolated compounds should be reported

Author Response

Dear Prof., Ph.D.,

I would like to thank the referees for your nice comments on our manuscript. I try my best to revise the manuscript with my co-authors. It spent me almost one week to contact with my co-authors and to discuss the improvement in both English grammar and contents of this manuscript. As you see, we made a major revision for this manuscript. You can find the modified and revised parts (marking out with red).

  1. We have carefully revised this paper, and supplemented and modified some contents.
  2. We have rewriteen and deleted some experimental contents in “Materials and Methods” part to make the experimental process more concise.

  3.  Yes, the comments are very nice. The structures of several MAAs have been added.

    We hope you could provide us more comments so that we can improve our future work. And I hope that our manuscript have chance to get a rapid publication in this Journal. Please understand my condition because the rapid publication of my work is important for me and our group. I am looking forward to your good news.

    Thank you for your attention and comments.

    Best Regards

    Yours sincerely

    Yingying Sun

Round 2

Reviewer 1 Report

The revised manuscript does not include improvement of the quality of the English.  It still needs a many corrections of the grammar and usage to bring it up to an acceptable standard of English. 

The authors have not sufficiently corrected the terminology for describing the extractions.  They use "extraction time" which I suspect refers to the duration of the extraction.  They also refer to "extraction times" which I guess might be the number of successive extractions performed.  The text needs to be clarified regarding these.  

The legends to Figures 1, 2, and 3 have very small squares that indicate which type of shading relates to each species.  These square are too small to determine what the type of shading is within them.  These boxes need to be increased in size.

It is not clear what the operating parameters of each of the MAA assays is - what is the linear range, what standards have been used to quantify and how were they obtained and their identify confirmed, what is the accuracy and precision.

The discussion does still does not include a clear description of the recommended procedure for extraction of MAAs from each species, and the recoveries of each of the MAAs when following a particular extraction method.  It would be very useful to know how much of each MAA present that is being recovered during the extraction, expressed as a percentage of the total present.  While it may not be possible to produce this data very easily, it should at least be discussed, as it is the major point of relevance in developing a method and recommending it to others.

Author Response

Dear Prof., Ph.D.,

I would like to thank you for your nice comments on our manuscript. We have carefully revised this manuscript according to your suggestions. I hope our revision can well express what we want to express, and correctly answer your questions. We made some modifications and additions for this manuscript. You can find the modified and revised parts (marking out with red).

  1. I try my best to revise the manuscript with my co-authors to improve the quality of English. Please review the revised manuscript.
  2. Yes, you are right. "extraction times" means “the number of successive extractions”. To resolve the ambiguity, we agreed to use extraction degree for extraction times.
  3. In order to make these boxes (the legends) in Figures 1, 2 and 3 more obvious, we use different colors and more obvious fill patterns.
  4. Your question is very professional and important. We have thought about the answer to this question for a long time. This is because in the process of identifying MAAs, we follow their HPLC-ESI-MS spectra, and comparison with existing literatures [6, 27, 28]. The conclusion obtained by such identification methods can only be "definitely" or "unknown", therefore, we do not know how to express this accuracy and precision. I hope to explain it to you clearly. I answered this question with great anxiety.
  5. Yes, this suggestion is very good. The objective of this article is to provide other researchers with the extraction methods and processes of MAAs from macroalgae. Therefore, in the discussion part, we clearly pointed out the specific extraction process parameters.

    In this study, we did not use MAAs contents as the measurement index to optimize the extraction process. Although we did not provide the exact content of MAAs in MAAs extracts from each marine macroalgae, but we gave their approximate values (now it has been added). Based on this, we compared MAAs contents in MAAs extracts from four species of marine macroalgae with that of other macroalgae. Please review the revised manuscript.

    I hope that our manuscript have chance to get a rapid publication in this Journal. Please understand my condition because the rapid publication of my work is important for our research term. I am looking forward to your good news.

    Thank you for your attention and comments.

    Best Regards

    Yours sincerely

    Yingying Sun

Reviewer 2 Report

The authors have responded adequately to my doubt, question and requests.

The paper have been improved. There is minor points t be clarified and correct 

Line 75 it is not correct to start the sentence with "And". it is necessary to check the manuscript because this mistake can be found through the paper. 

Line 188 Yield instead of "Yeilds"

Line 191 MAA extracts instead of "MAAs extrcts"

It is necessary to clarify if biomass is expressed in g Fresh Weight (FW) or dry weight (DW)

Line 341 Porphyra leucosticta instead of "Porphyra leucosticte"

Author Response

Dear Prof., Ph.D.,

Thank you for your valuable suggestions for this manuscript. According to these suggestions, we have modified the corresponding parts. You can find the modified and revised parts (marking out with red).

 Line75~Line 341, Your suggestion is very correct. In the revised manuscript, corresponding modifications have been made.

Thank you for your attention and comments.

Best Regards

Yours sincerely

Yingying Sun

Reviewer 3 Report

The major points raise by the reviewers have been addressed

Author Response

Dear Prof., Ph.D.,

Thank you very much for your affirmation of our revision work. And thank you again for your valuable suggestions for this manuscript.  We made some modifications and additions for this manuscript. You can find the modified and revised parts (marking out with red).

Best Regards

Yours sincerely

Yingying Sun